# Characteristics and Outcomes of Critically Ill Pregnant/Postpartum Women with COVID-19 Pneumonia in Western Balkans, The Republic of Srpska Report

**DOI:** 10.3390/medicina58121730

**Published:** 2022-11-26

**Authors:** Pedja Kovacevic, Sandra Topolovac, Sasa Dragic, Milka Jandric, Danica Momcicevic, Biljana Zlojutro, Tijana Kovacevic, Dragana Loncar-Stojiljkovic, Vlado Djajic, Ranko Skrbic, Vesna Ećim-Zlojutro

**Affiliations:** 1Medical Intensive Care Unit, University Clinical Centre of The Republic of Srpska, 78000 Banja Luka, Bosnia and Herzegovina; 2Faculty of Medicine, University of Banja Luka, 78000 Banja Luka, Bosnia and Herzegovina

**Keywords:** COVID-19, critical illness, pregnant/postpartum women, outcomes

## Abstract

*Background and Objectives*: Coronavirus disease 2019 (COVID-19) is a novel infectious disease that has spread worldwide. As of 5 March 2020, the COVID-19 pandemic has resulted in approximately 111,767 cases and 6338 deaths in the Republic of Srpska and 375,554 cases and 15,718 deaths in Bosnia and Herzegovina. Our objective in the present study was to determine the characteristics and outcomes of critically ill pregnant/postpartum women with COVID-19 in the Republic of Srpska. *Materials and Methods*: The retrospective observational study of prospectively collected data included all critically ill pregnant/postpartum women with COVID-19 in a university-affiliated hospital between 1 April 2020 and 1 April 2022. Infection was confirmed by real-time reverse transcriptase polymerase chain reaction (RT-PCR) from nasopharyngeal swab specimens and respiratory secretions. Patients’ demographics, clinical and laboratory data, pharmacotherapy, and neonatal outcomes were analysed. *Results*: Out of the 153 registered pregnant women with COVID-19 treated at the gynaecology department of the University Clinical Centre of the Republic of Srpska, 19 (12.41%) critically ill pregnant/postpartum women (median age of 36 (IQR, 29–38) years) were admitted to the medical intensive care unit (MICU). The mortality rate was 21.05% (four patients) during the study period. Of all patients (19), 14 gave birth (73.68%), and 4 (21.05%) were treated with veno-venous extracorporeal membrane oxygenation (vvECMO). *Conclusions*: Fourteen infants were born prematurely and none of them died during hospitalisation. A high mortality rate was detected among the critically ill pregnant/postpartum patients treated with mechanical ventilation and vvECMO in the MICU. The preterm birth rate was high in patients who required a higher level of life support (vvECMO and ventilatory support).

## 1. Introduction

Coronavirus disease 2019 (COVID-19) is a novel infectious disease that affects the respiratory system. It was first reported in Wuhan China, and it has subsequently spread worldwide [1,2]. As of 5 March 2020, the COVID-19 pandemic has resulted in approximately 111,767 cases and 6338 deaths in the Republic of Srpska and 375,554 cases and 15,718 deaths in Bosnia and Herzegovina [3,4,5]. Data from viral respiratory illnesses such as influenza, severe acute respiratory syndrome corona virus 1, and Middle East respiratory syndrome suggest that viral respiratory infection during pregnancy may worsen both maternal and fetal outcomes [6]. Certain physiological characteristics of the immune and respiratory systems may make pregnant women with COVID-19 more susceptible to critical illness [7]. Some studies suggest that among adolescents and women aged 15–44 years with COVID-19, pregnancy is associated with an increased risk of ICU admission and the need for mechanical ventilation [8]. A worrying fact in the treatment of critically ill COVID-19 pregnant women in the Republic of Srpska is that, by April 2022, the majority of COVID-19-related deaths occurred in low- and middle-income countries (LMICs). The Republic of Srpska, as part of Bosnia and Herzegovina, belongs to countries called low-resource settings, which is quite similar to LMICs. Generally, and, thus, in the pre-pandemic period, the burden of critical illness in LMICs is substantial, and mortality rates remain unacceptably high when compared to those of high-income countries (HIC). While data on the structure, treatment, and outcomes in LMICs are scarce, data from HICs cannot be easily extrapolated [9,10]. Globally, there are over 213 million pregnancies every year, of which an estimated 190 million (89%) occur in low-resource settings, where the risk of poor birth outcomes is the highest. Currently, the World Bank classifies Bosnia and Herzegovina (as well as the Republic of Srpska) as an upper-middle-income country, but the healthcare systems in such countries are often underdeveloped and quite similar to the ones in LMICs (hence, why they are often referred to as low-resource countries) [9]. Data describing the characteristics and treatment outcomes of critically ill pregnant/postpartum women with COVID-19 in low-resource settings are very scarce in the literature. Our objective in the present study was to determine characteristics and outcomes of critically ill pregnant/postpartum women with COVID-19 in the Republic of Srpska.

## 2. Materials and Methods

### 2.1. Study Design

A retrospective observational study was conducted in the 28-bed medical intensive care unit (MICU) of university-affiliated, tertiary-care hospital: the University Clinical Centre of the Republic of Srpska (UCC RS), Banja Luka, between 1 April 2020 and 1 April 2022. MICU in UCC RS has been established during the last 15 years with the support of critical care specialists trained in the United States and Europe, members of the European Society of Intensive Care Medicine, and the Society of Critical Care Medicine. The MICU serves as a referral Centre for the region of the Republic of Srpska, with approximately 1,000,000 inhabitants, and is currently the most advanced multidisciplinary MICU in Bosnia and Herzegovina, and, therefore, in the Republic of Srpska [10,11,12,13]. Since the establishment of the MICU in Banja Luka, critically ill patients, including pregnant women, from the whole of the Republic of Srpska are transferred to UCC RS’s MICU. It should be noted that Bosnia and Herzegovina is in a post-war period, is a country in transition, and is part of a region known as the Western Balkans. The study protocol was approved by the Ethics Committee of the UCC RS, No. 01-19-155-2/22.

### 2.2. Patients’ Selection

The study included all critically ill and mechanically ventilated patients with confirmed SARS-CoV-2 virus infection and acute respiratory distress syndrome ARDS [14]. Reverse transcription polymerase chain reaction (RT-PCR) from nasopharyngeal swab specimens and respiratory secretions was taken ≤14 days prior to hospital admission. Inclusion criteria for participation in this study were: women in pregnancy/postpartum with signs of severe respiratory insufficiency and in need of treatment in MICU. Detailed admission criteria for MICU are listed below (Section 2.3). Exclusion criteria were not defined. Patients were categorised based on the most invasive respiratory support required during the course of the treatment. Lowest to highest severity was defined as follows: group A—requirement of oxygen insufflation (e.g., by nasal prongs) and high-flow nasal oxygen therapy (patients with P/F ratio > 200) and patients with mild ARDS according to Berlin criteria for ARDS) and group B—non-invasive ventilation (NIV), invasive ventilation through an endotracheal tube, and veno-venous extracorporeal membrane oxygenation (vvECMO) (patients with P/F ratio < 200 and patients with moderate and severe ARDS according to Berlin criteria for ARDS).

### 2.3. Criteria for MICU Admission

Admission or transfer to the MICU (level 3) was indicated by shared decision between gynaecologists and intensivists. Criteria for MICU admission were severe hypoxic respiratory failure, severe multi-organ involvement, the need for renal replacement therapy, or vasopressor/inotropic support. Severe hypoxic respiratory failure was defined as a blood oxygen saturation level (SpO_2_) < 94% despite maximal oxygen flow rate and/or a respiratory rate ≥ 20/min and/or a partial pressure of oxygen in blood (PaO_2_) < 70 mmHg, a partial pressure of carbon dioxide in blood (PaCO_2_) <30 or >45 mmHg, or a pH < 7.3 in arterial blood gas. Foetal monitoring was performed by regular ultrasound assessment of foetal growth and Dopplers and daily non-stress test (foetal heart-rate monitoring) after 24 weeks of gestation. Indications for delivery were: inability to reach mechanical ventilation targets to ensure appropriate transplacental gas exchange, driving pressures >15 cmH_2_O, a ratio of arterial oxygen partial pressure (PaO_2_ in mmHg) to fractional inspired oxygen (P/F ratio) ≤150 mmHg despite prone positioning, or other obstetric indications.

### 2.4. Study Parameters

Primary outcomes were maternal mortality (death during pregnancy or after the end of pregnancy), preterm delivery (delivery before 37 completed weeks of gestation), and neonatal mortality (stillbirth). We collected information on pregestational chronic diseases, gestational age at the beginning of COVID-19 in patient, and clinical care data such as induction of labor and mode of delivery. The following data were collected as well: demographics, laboratory parameters, radiological information, and mechanical ventilation details. Severity of illness was assessed according to the Simplified Acute Physiology Score (SAPS) II score, Acute Physiology and Chronic Health Evaluation (APACHE) II score, and Sequential Organ Failure Assessment (SOFA) [15,16]. Information regarding the need for vasopressor treatment, mechanical ventilation, renal replacement therapy, prone position, or vvECMO was also recorded, and patients’ vaccinal status was assessed. The presence of ARDS was classified according to Berlin ARDS criteria: no, mild, moderate, and severe ARDS [14].

### 2.5. MICU Treatment of Critically Ill Patients in MICU of the University Clinical Centre of the Republic of Srpska

Mechanical ventilation: for patients from group B (moderate-severe ARDS—P/F ratio ≤ 200 mmHg), treatment consisted of prone positioning (postpartum), low tidal volumes, positive end-expiratory pressure (PEEP) titration according to the higher PEEP table, lower fraction of inspired oxygen, and restrictive fluid management. In a case of ongoing pregnancy, women (four patients from group A) were managed in 15° left lateral position when in dorsal decubitus, and we aimed at maintaining maternal SpO_2_ ≥ 94%, an arterial pH ≥ 7,25, PaCO_2_ between 30–60 mmHg, and PaO_2_ ≥ 70 mmHg to ensure appropriate transplacental gas exchange for the foetus. The protocol for mechanical ventilation and treatment of respiratory failure in our centre was similar to the therapeutic approach in other studies [17].

(a)Corticosteroids: All women requiring oxygen therapy were treated with corticosteroids. Intravenous methylprednisolone, to limit transplacental passage, was administered in a dose of 1 mg/kg daily for 10 consecutive days, or for a shorter period if oxygen therapy was no longer required. In case foetal lung maturation was required between 24 and 34 weeks of gestation, methylprednisolone was replaced by 6 mg of dexamethasone intravenously twice a day for 48 h. In case of moderate/severe ARDS (P/F ratio < 200 mmHg) and poor response to prone positioning or progression of disease on chest computed tomography (Group B), higher dose of methylprednisolone was administered (2 mg/kg) intravenously. Progression on CT was reported in presence of bilateral ground-glass opacity and/or consolidation, which was in the line with other clinical recommendations, and CT scans for all patients were performed before transfer to MICU [18,19,20].(b)Anticoagulation: All patients were on therapeutic doses of heparin, except for patients with contraindication for this medication. Unfractionated heparin was dosed in accordance with levels of Activated Partial Thromboplastin Clotting Time (aPTT). None of observed patients experienced thromboembolic or bleeding events. This is in contrary to other clinical approaches [21,22].(c)Veno-venous extracorporeal membrane oxygenation (vvECMO) was applied to four patients (21.05%); this very sophisticated and very challenging therapeutic mode was accessible to all patients from this population (pregnant/postpartal women with COVID-19).(d)Specific medication: None of our patients received antiviral therapy, tocilizumab, or orcasirivimab/imdevimab. Some centres had experience with mentioned medications [23].

### 2.6. Statistical Analysis

Data are reported as means and standard deviations for normally distributed continuous data, and medians and interquartile range (IQR, 25–75) are not for normally distributed continuous data or as absolute numbers and percentages of categorical data. Normality of the data was tested using Kolmogorov–Smirnov test. Two-sided statistical significance was set at *p* < 0.05. Kruskal–Wallis, Mann–Whitney U, and Student’s t-tests were applied for continuous data. We performed Pearson chi-squared test of independence to compare frequencies of pre-existing medical conditions, renal replacement therapy, vasoactive drugs, corticosteroids, and therapeutic doses of anticoagulants between the two groups with different forms of respiratory support. Descriptive data analysis, statistical analysis, and visual presentation were performed with SPSS 26 (IBM Corp. Released 2019. IBM SPSS Statistics for Windows, Version 26.0. Armonk, NY, USA: IBM Corp).

## 3. Results

The number of pregnant women that were admitted to the gynaecology department of UCC RS and diagnosed with COVID-19 during the study period was 153. Nineteen pregnant/postpartum women who were critically ill and required treatment in MICU were found to be eligible for this study, which is 12.41% of the total registered pregnant women.

### 3.1. Maternal and Neonatal Outcomes

The MICU mortality of pregnant/postpartum women critically ill with COVID-19 in the study period was 21.05% (four patients). All patients who died were in defined group B (21.05%) vs. group A (0%) *p* < 0.001, and the length of the MICU stay was significantly higher in group B as well (*p* < 0.05). Of all patients (19), 14 gave birth during ICU stay (73.68%) and distribution of mode of delivery was as follows: 9 primary caesarean sections and 5 emergency caesarean sections. All patients were included in the study before the delivery. Patients were delivered during the first 24–48 h of ICU stay, or the pregnancy continued beyond ICU discharge. The main reason for the initiation of caesarean section was the extremely severe respiratory insufficiency and critically ill condition of the pregnant women. In four pregnancies, the mode of delivery was not reported due to an ongoing pregnancy. Five (35.71%) patients received general anesthesia with invasive ventilation during delivery, and nine (64.28%) received spinal anesthesia. Four patients (21.05%) were treated with vvECMO. One pregnant woman was admitted to the MICU in an extremely critical condition and was resuscitated continuously; unfortunately, she died despite the provision of continuous advanced life support for several hours. All patients except one were in the third trimester of pregnancy. Fourteen infants were born prematurely (i.e., delivery before completion of 37 gestational weeks), and all of them required admission to the neonatal ICU (e.g., prematurity-related problems and respiratory maladaptation). Swabs were taken from all live-born neonates for PCR testing for SARS-CoV-2 RNA at birth, and none of them were positive. The maternal and neonatal outcomes for each type of COVID-19 treatment are shown in Table 1.

### 3.2. Basic Characteristics of Patients (Maternal) and Neonates

Patients had a median age of 36 (IQR, 29–38) years. Table 2 shows an overview of the maternal demographics and clinical characteristics in groups A and B and the basic characteristics of the neonates, while the laboratory characteristics for both groups of the studied population are shown in Table 3.

The mean ± SD of the gestational age of all the born babies in this study was 31.43 ± 4.35. The main clinical symptoms on admission to the MICU were dyspnoea, coughing, and fever, which were present in 16 of 19 patients (84.21%). All 19 patients had bilateral infiltrates seen on a chest X-ray (CXR) or computerised tomography (CT) scan, along with a certain degree of ARDS. The most common findings during examination of the chest CT were bilateral diffuse ground glass-like infiltrations and pneumonic consolidation areas. On the chest X-rays, diffuse bilateral infiltration and increased opacity were seen (Figure 1 and Figure 2).

None of the pregnant women observed in this study had severe respiratory disease pre-pregnancy or family genetic disorders, nor any other significant comorbidities including obesity. Sources in the literature show a lack of monitoring these parameters in similar patients during the pandemic period in the Balkans (especially in the Western Balkan). Unfortunately, none of the hospitalised pregnant COVID-19 patients (critically ill or non-critically Ill) had received any type of COVID-19 vaccination.

### 3.3. Characteristics of Treatment in MICU

Generally, admission to the MICU occurred within 10 days of a SARS-CoV-2 infection diagnosis. The median length of MICU stay was 5 days (IQR, 3–12). Differences between the forms of respiratory support (for groups A and B) regarding patients’ characteristics and treatment modalities are shown in Table 2 and Table 3. Blood samples for all laboratory findings are taken on admission. All postpartum patients (14 patients, 73.68%) were in a prone position, regardless of whether they were on mechanical ventilation, on vvECMO, or awake. The prone position was applied for all patients immediately after delivery. All observed patients received anticoagulants (therapeutic doses) and no thromboembolic events or major bleeding were reported. Follow-up of all recovered patients after MICU discharge showed that there were no specific events.

## 4. Discussion

Our study revealed that during the pandemic period, out of 153 registered pregnant women with COVID-19, 19 (12.41%) critically ill pregnant/postpartum women were admitted to the MICU. This percentage is similar to previously reported rates [25,26]. It is well-known that admission to the MICU is associated with an elevated risk for a poor maternal and foetal outcome generally, and recent studies have reported that pregnancy is an independent risk factor for critical illness regarding COVID-19 [27]. Sources in the literature (data) for the population of pregnant and postpartum women with severe COVID-19 are scarce, and knowledge is limited (especially in LIMICs and in low-resource settings generally). Due to many differences in healthcare systems, a regional (area of the Balkans—Southeast Europe) analysis may be helpful. To the best of our knowledge, our study is the only one conducted in the area of the Balkans that determines the characteristics and outcomes of critically ill pregnant/postpartum women with COVID-19. The most dominant symptoms in all patients in the MICU were cough, fever, and dyspnoea, and these symptoms are also described in a large number of the studies that have followed pregnant/postpartum COVID-19 patients [28,29]. The MICU mortality of this very sensitive group of patients was 21.05% (four patients) during the observed pandemic period, and the main cause of death was severe respiratory insufficiency (ARDS), refractory to all therapeutic options. These results are in accordance with results from other similar countries (LIMICs) and even from some developed countries (i.e., Turkey and Saudi Arabia) [25,26]. On the other hand, in HICs, the mortality of critically ill pregnant/postpartum women is lower (about 5–10%) [30,31]. Out of a total of 19 patients, 11 (57.89%) required a lower level of life support (oxygen insufflation and high-flow oxygen therapy) during their stay at the MICU, and they had mild or moderate ARDS (Group A). This is similar to other reports from LIMICs, in contrast to HICs, where the percentage of patients who require a lower level of life support in the MICU is smaller [30,32]. The organisation and existence of a large number of COVID-19 high dependency units (HDU) in HICs may provide an explanation for the facts mentioned above [31]. Pregnant/postpartum women who were on a higher level of life support (group B) were older than the patients who required a lower level of life support (Group A), but this difference was not statistically significant.

Patients from group B who were on a higher level of life support had a significantly increased mortality rate and length of MICU stay. Similar studies reported higher rates of maternal mortality, morbidity, and early caesarean-section delivery primarily due to the mother’s deteriorating health or foetal distress [33,34,35]. All patients who required higher level of life support (group B) had significantly higher values of SAPS II and APACHE II (ICU scoring systems) than the patients from group A. These patients (group B) were significantly more often treated with continuous renal replacement therapy (CRRT). The use of vasopressors in patients who required a higher level of life support was significantly higher than in patients who required a lower level of life support. Similar results were observed in previous studies [30,31].

The laboratory findings showed significantly higher levels of procalcitonin and blood urea nitrogen (BUN). The levels of C-reactive protein (CRP), total white blood cell count, and D-dimer were higher in severe cases (Group B), but without statistical significance, while postpartum status itself can contribute to elevated CRP and D-dimer levels in addition to the severity of COVID-19 infection. In group B, haemoglobin level and platelet count were lower, also without reaching statistical significance. The literature describes conflicting results, but most studies indicate that raised CRP and total white blood cell count were reliable indicators of severity in COVID-19 pregnant/postpartum women [36]. Significantly higher levels of procalcitonin in the group of patients who required a higher level of life support and more invasive procedures can be associated with a higher incidence of intrahospital infection in COVID-19 patients; this was actually shown in many studies [37,38]. All patients in our study who were intubated and on vvECMO life support developed some kind of intrahospital bacterial infection, and none of them had a fungal infection. The problem of superinfections in critically ill COVID-19 patients is well-recognised generally, but it is probably much more prominent in low-resource settings.

It was evident in our study that neonates born to critically ill COVID-19 patients—group B had a significantly higher chance of being born prematurely than those born to mothers with a milder COVID-19 disease—group A (87.50% vs. 63.63%; *p* < 0.05). Fourteen infants were born prematurely, and all of them were treated in the neonatal ICU. The gestational age (mean) and Apgar in 1. and 5. minutes (median) scores were not significantly different between newborns from the different groups (group A vs. group B). On the other hand, newborns from mothers who required a higher level of life support (group B) had a significantly lower birth weight. After treatment at the neonatal ICU, all newborns were discharged. The high survival rate of these newborns is in line with HICs’ results [30,31]. Finally, one very important finding of this study was the vaccinal status of the pregnant population. Exactly none of the observed patients in this study were vaccinated against COVID-19. To the best of our knowledge, this is the only study that shows the rate of vaccination of pregnant/postpartum women in this region (Southeast Europe). It is well-known that vaccination against COVID-19 can reduce critical illness and fatal ARDS in pregnant/postpartum women as well as protect their offspring, but their hesitancy to receive COVID-19 vaccination is very high and worrying, even in HICs [39,40,41,42]. None of the pregnant patients were prone before delivery.

The most important limitations of the study were the small number of patients, the single-centre design, and the fact that none of the included pregnant/postpartum women had comorbidities. On the other hand, the very small, almost negligible amount of scientific research in low-resource countries and the related writing about the treatment outcomes in these settings can be an invisible limitation. Especially because this type of research can sometimes be accompanied by political and public impact, treatment outcomes in these countries are often much worse compared to those in HICs. The main strength of this study is that it showed the outcomes of the treatment of critically ill pregnant/postpartum women in the Western Balkans. Any study published in low-resource countries and LMICs can stimulate other researchers from these settings to publish their results, which is an invisible strength as well.

## 5. Conclusions

In conclusion, critical COVID-19 illness in pregnant/postpartum patients who require high life support measures (invasive mechanical ventilation and vvECMO) was connected with a high mortality rate, which is in accordance with the global experience. This population of patients is at high risk for increased hospital stays and caesarean-section delivery due to COVID-19’s progression. The fatality rate of pregnant/postpartum women is much more prominent in low-resource countries (settings). The neonates born to such mothers may be premature and often need treatment in a neonatal ICU. Nevertheless, adequate treatment with respiratory support and interdisciplinary management of critically ill pregnant/postpartum COVID-19 patients may not only lead to maternal recovery but also to an improvement in neonatal outcomes. The establishment and development of a modern multidisciplinary ICU has had a tremendous effect on the global health system within one country. All observed patents did not have any additional illness including obesity. Hence, the risk factors that influenced the final outcome in these patients were almost negligible. Highlighting the importance of vaccination among pregnant women may contribute to increasing COVID-19 vaccination rates and to overcoming vaccine hesitancy. Further research and reports of treatment of critically ill pregnant/postpartum women infected with COVID-19 is necessary.

## Figures and Tables

**Figure 1 medicina-58-01730-f001:**
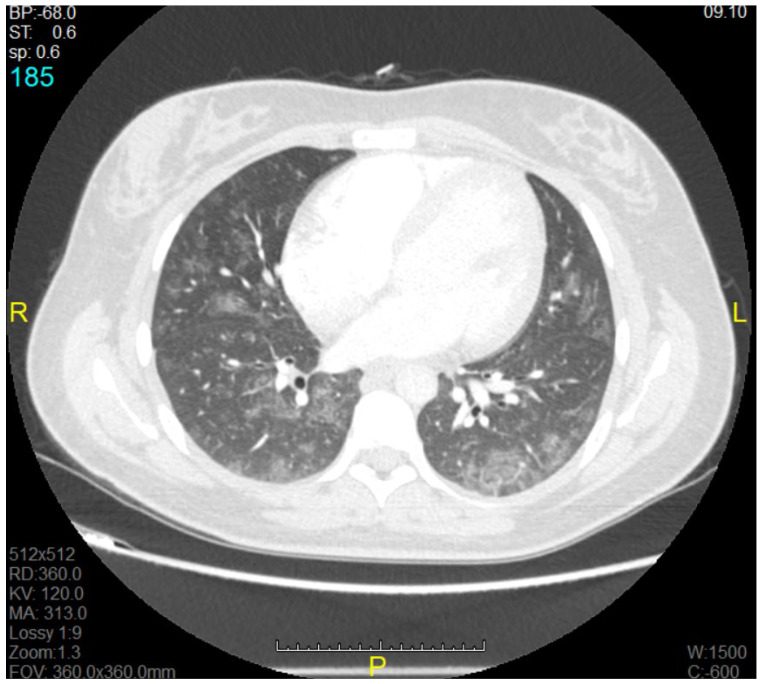
Computerised tomography (CT) image of a pregnant woman with acute respiratory distress syndrome (ARDS).

**Figure 2 medicina-58-01730-f002:**
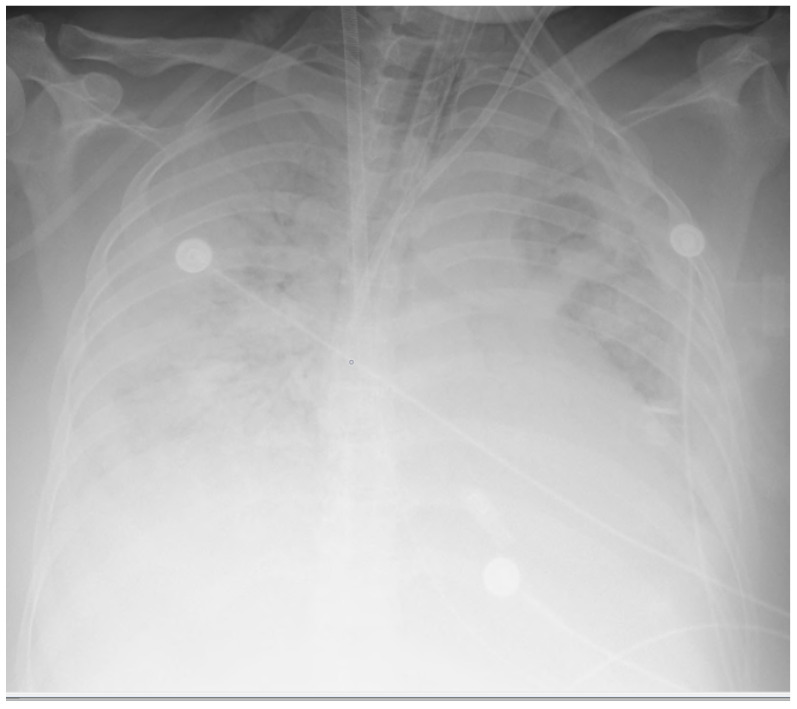
Chest X-ray showing diffuse opacity and infiltrates in a postpartum woman treated with vvECMO.

**Table 1 medicina-58-01730-t001:** Final maternal and neonatal outcomes in groups with different types of respiratory support (expressed in absolute numbers): O_2_—patient who required oxygen insufflation; HFNC—patient who required high-flow oxygen therapy; NIV—noninvasive mechanical ventilation; IMV—invasive mechanical ventilation; vvECMO—veno-venous extracorporeal membrane oxygenation.

Groups	Mild ARDS (O_2_ + HFNC) N = 11	Moderate ARDS (NIV) N = 8	Severe ARDS (IMV)	Severe ARDS (vvECMO)	All
Recovery	11	2	1	1	15
Death	0	0	1	3	4
Labor (preterm)	7	2	1	4	14
Ongoing pregnancy	4	0	0	0	4
Livebirth	7	2	1	4	14

**Table 2 medicina-58-01730-t002:** Demographic and clinical characteristics of all treated pregnant/postpartum women and basic characteristics of neonates.

Groups	O_2_ + HFNC (Group A), N = 11 (57.89%)	NIV + IMV + ECMO (Group B), N = 8 (42.10%)	*p*
Maternal age, median (IQR)	31 (25–37)	36 (35–40.75)	0.199
Gestational age at diagnosis (weeks), (mean ± SD)	32.86 ± 4.56	30.00 ± 3.92	0.233
Birth weight in grams (mean ± SD)	2758.57 ± 669.36	1664.29 ± 634.16	0.009
Apgar in 1 minute (median, IQR)	9 (7–9)	8 (6–8)	0.316
Apgar in 5 minutes (median, IQR)	9 (9–9)	8 (8–9)	0.259
Length of ICU stay (days), median (IQR)	4 (3–5)	13.5 (8.5–20.5)	0.015
Renal replacement therapy	0	3 (100%)	0.021
Vasoactive substances	0	6 (75%)	0.001
Thromboembolic events	0	0	-
Anticoagulation:			
Prophylactic	0	0	-
Therapeutic	11 (100%)	8 (100%)	-
Steroids	11 (100%)	8 (100%)	-
Coexisting conditions:	3 (27.27%)	3 (37.50%)	0.636
COVID-19 vaccination	0	0	-
Severity of illness:			
SAPS II	6 (6–7)	16.5 (8.25–20.75)	0.007
APACHE II	4 (4–5)	10 (4.75–10-75)	0.012
SOFA on admission	4 (4–4)	4 (3–4.75)	0.904
*P/F ratio*	>200	<200	-

O_2_—patient who required oxygen insufflation; HFNC—patient who required high-flow oxygen therapy; NIV—noninvasive mechanical ventilation; IMV—invasive mechanical ventilation; ECMO—extracorporeal membrane oxygenation; N—number of patients; SAPS II—Simplified Acute Physiology Score; APACHE II—Acute Physiology and Chronic Health Evaluation; SOFA—Sequential Organ Failure Assessment. Laboratory analysis was done at the time of admission.

**Table 3 medicina-58-01730-t003:** Laboratory parameters of all treated pregnant/postpartum women.

Parameters	Normal Values	Pregnancy Effect *	O_2_ + HFNC (Group A), N = 11 (57.9%)	NIV + IMV + ECMO (Group B) N = 8 (42.10%)	*p*
CRP, mean ± SD	0–5 mg/L	0.4–8.1 (third trimester)	88.38 ± 64.53	104.30 ± 31.56	0.530
Haemoglobin levels, mean ± SD	138–175 g/L	decreased for 14–20 g/L	109.45 ± 12.03	105.50 ± 12.58	0.501
Platelet count, mean ± SD	158–424 × 10^9^/L	Slight decrease	241.36 ± 104.15	189.13 ± 69.19	0.235
White blood count, median (IQR)	3.4–9.7 × 10^9^/L	Increased for 3.5	9.36 (7.57–14.11)	7.04 (5.30–10.40)	0.137
D-dimer, median (IQR)	0–0.05 mg/L	0.13–1.7 (third trimester)	1.53 (0.95–5.45)	1.84 (1.21–2.11)	0.804
BUN, median (IQR)	2.8–7.2 mmol/L	1.1–3.9 (third trimester)	48.5 (46–51.5)	58 (54–62)	0.010
PCT, median (IQR)	<0.02 ng/mL	-	0.09 (0.05–0.20)	0.26 (0.16–0.51)	0.047

O_2_—patient who required oxygen insufflation; HFNC—patient who required high-flow oxygen therapy; NIV—noninvasive mechanical ventilation; IMV—invasive mechanical ventilation; ECMO—extracorporeal membrane oxygenation; N—number of patients; CRP—C-reactive protein; BUN—blood urea nitrogen; PCT—procalcitonine. * Reference for corrected laboratory findings: Abbassi-Ghanavati M, Greer LG, Cunningham FG. Pregnancy and laboratory studies: a reference table for clinicians [24].

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
