# Peer review of "Characteristics and Outcomes of Critically Ill Pregnant/Postpartum Women with COVID-19 Pneumonia in Western Balkans, The Republic of Srpska Report"

_medicina, 2022, doi:10.3390/medicina58121730_

Round 1
Reviewer 1 Report
Line 45: "The worrying fact in the treatment of critically ill COVID-19 pregnant women in the Republic of Srpska is that by April 2022, the majority of COVID-19–related deaths occurred in low- and middle-income countries (LMICs)."
-Rewrite this statement for clarity. Does the author mean the Republic of Srpska is a low-to-middle-income entity?
-Line 55-56: Provide a reference for this statement.
-What is the power or alpha of the study which helps define the validity of the results? Also, no exclusion criteria were defined. One important limitation of the study is it did not clarify if the study population had any associated comorbid condition/conditions. Hence, results can be biased. Not all fatal outcomes are due to COVID-19, and pregnant or post-partum women with co-morbidities such as preeclampsia/eclampsia, gestational diabetes, and obesity may have played an important role in deciding the outcome. This is very important as these risk factors are well-known, leading to worse outcomes in COVID-19 infection, just like vaccination status, and should be clarified.
-Line 283: The authors should mention that post-partum status itself can contribute to elevated CRP and d-dimer levels in addition to the severity of COVID-19 infection.
-Overall, interesting article that looks into the severity of COVID-19 infection in a pregnant or post-partum population.
Reviewer 2 Report
I appreciate the efforts of the authors in collating the data of covid pregnant patients
It looks just an audit of pregnant patients with covid admitted in hospital. Over last two years of pandemic.
Numbers are only 19 – not a big number to conclude except that the services provided by the hospital looks quite decent as far as outcome is concerned
Vaccination status of those who were not admitted in ICU ?
None of icu patients were vaccinated – agreed but were those not admitted in icu were vaccinated?
criteria for labeling severe or critical not very clear - what was pf ration or what Fio2 was when you decided that SPo2 less than 94 or Pao2 less that 70 - need a bit more clarification
Table 1
What was the P/F ratio of the patients in each group? The description is incomplete without mentioning the severity of ARDS and justifying the O2 support provided for the patients.
Gestational age depicted in Table 1 more than 30 weeks – it seems all were in third trimester
Fig 1 This graph is very confusing. Please simplify it so that overall outcome is clear based on the oxygen support required.
Table 2
At what time these lab values taken – at admission or worst values?
Normal ranges? Are these reference values for pregnant or non-pregnant population? All these parameters differ for pregnant patients. Please post a reference for the reference values.
Higher urea in less sicker patients. What is the significance of this p-value. Patients who are sicker (i.e. requiring IMV or ECMO) generally will have higher urea or creatinine values. Here we are seeing opposite to this..
PCT for patients of IMV or ECMO will obviously be higher when compared to those managed non-invasively of high flow oxygen.
How many developed secondary bacterial or fungal infection? What were the outcomes in those developing secondary infection? – are these at the time of admission?
Were any specific changes noted in pregnant patients. A CT scan doesn't change the outcome and at our institute we don't routinely do a CT chest for pregnant patients. Was there a specific indication to do a CT for this patient? Did this patient require IMV? What was the outcome of this patient?
Fig 3 Cxray - What was the outcome of this patient?
Page 7 section 3.3 lines 231
1. The length of ICU stay seems to be less. What was the length of hospital stay? In conclusion it has been stated that pregnant patients tend to have a longer hospital stay. Please support the statement with data.
2. Prone positioning after delivery - What if the patient needed prone positioning before delivery?
3. Therapeutic doses of which anticoagulant?
4. Was any of the patients positioned lateral? Was there any benefit of lateral positioning when prone positioning could not be performed?
5. For how long the patients were followed? Were there no respiratory symptoms?
Page 8 Line 286 Significantly higher levels of procalcitonin 286 in group of patients who require higher level of life support and more invasive procedures 287 can be in associated with higher incidence of intrahospital infection in COVID 19 patients 288 what was actually showed in many studies [35, 36].
Page 9 line 337
This is a retrospective study. How was the consent obtained from the patients after discharge. How was the consent obtained from the patients who did not survive?
Round 2
Reviewer 1 Report
The authors have tried to incorporate reviewer suggestions to the best of their ability.
The study's major limitation is the small sample size and no information on associated co-comorbidities or known risk factors such as obesity to increase covid-19 related mortality/morbidity. Hence, the results and conclusion cannot be relied upon.
Reviewer 2 Report
Authors have have made few changes and also replied to the queries raised.
Query ----Fig 1 This graph is very confusing. Please simplify it so that overall outcome is clear based on the oxygen support required.
Reply Thank you very much for your suggestion, I created completely new table (now this is table 1), and now in article exists tree tables and two figures
Comments - Authors have have made few changes and also replied to the queries raised. The table 1 which is being provided by the authors in place of the Fig is Ok but there is still need to provide information on outcome of patients based on the admission status ie severity status at the time of admission and the ultimate outcome rather than the ultimate treatment modality and the outcome ( as is shown in the table)
Query
--- Table 2
At what time these lab values taken – at admission or worst values?
Reply - All presented laboratory findingswere taken at admission line
Comment - Please mention in the table legend – lab investigations at the time of admission
Query
2. Prone positioning after delivery - What if the patient needed prone positioning before delivery?
Reply In case of ongoing pregnancy, women (four patients from group A) were managed in 15°
left lateralposition when in dorsal decubitus and we aimed at maintaining maternal SpO2
≥ 94%, an arterialpH ≥7,25, PaCO2 between 30-60 mmHg, and PaO2 ≥ 70mmHg to
ensure appropriate transplacentalgas exchange for the fetus.
Comment - So no one was proned before delivery. Authors Need to state explicitly the same .
